# Epigenetic Dysregulation in the Pathogenesis of Systemic Lupus Erythematosus

**DOI:** 10.3390/ijms25021019

**Published:** 2024-01-13

**Authors:** Yasuto Araki, Toshihide Mimura

**Affiliations:** Department of Rheumatology and Applied Immunology, Faculty of Medicine, Saitama Medical University, Saitama 350-0495, Japan; toshim@saitama-med.ac.jp

**Keywords:** systemic lupus erythematosus, epigenetics, chromatin, DNA methylation, histone modification, microRNA

## Abstract

Systemic lupus erythematosus (SLE) is a multisystem autoimmune disease in which immune disorders lead to autoreactive immune responses and cause inflammation and tissue damage. Genetic and environmental factors have been shown to trigger SLE. Recent evidence has also demonstrated that epigenetic factors contribute to the pathogenesis of SLE. Epigenetic mechanisms play an important role in modulating the chromatin structure and regulating gene transcription. Dysregulated epigenetic changes can alter gene expression and impair cellular functions in immune cells, resulting in autoreactive immune responses. Therefore, elucidating the dysregulated epigenetic mechanisms in the immune system is crucial for understanding the pathogenesis of SLE. In this paper, we review the important roles of epigenetic disorders in the pathogenesis of SLE.

## 1. Introduction

Systemic lupus erythematosus (SLE) is a chronic prototypic autoimmune disease that results from immune system-mediated inflammation and tissue damage [1]. Aberrant activation of the immune system leads to the production of a broad array of autoantibodies specific for nucleic acids and nucleic acid-binding proteins, including anti-nuclear antibodies (ANAs), anti-double stranded DNA (dsDNA) antibodies, and anti-Smith (Sm) antibodies [2]. The autoreactive immune responses result in several characteristic clinical features, including skin rashes, oral ulcers, inflammatory polyarthritis, serositis, neuropsychiatric disorders, glomerulonephritis, and blood cell abnormalities [3]. SLE is a heterogeneous disease with varying combinations of clinical features and is characterized by a relapsing and remitting clinical course and a highly variable prognosis. SLE primarily affects females of childbearing age, with the highest prevalence in African-American, Asian, and Hispanic populations [4]. Therapy for SLE includes corticosteroids, hydroxychloroquine, immunosuppressants, and biologic agents targeting specific molecular mechanisms such as B cell-activating factor belonging to the tumor necrosis factor family (BAFF, also called BLyS) and type I interferon (IFN) receptor subunit 1 [5].

A line of evidence has demonstrated that a variety of environmental factors trigger autoimmune diseases such as SLE in genetically predisposed individuals [6,7]. SLE exhibits a strong familial accumulation with a much higher frequency among first-degree relatives of patients. SLE develops concordantly in approximately 25–50% of monozygotic twins and 5% of dizygotic twins. In spite of the influence of heredity, most cases appear sporadically. Genetic and environmental factors are involved in the immune system dysregulation that can trigger the development of SLE [8]. Epigenetic mechanisms, including DNA methylation, histone modifications, and microRNA (miRNA) expression, have also been shown to be associated with the pathogenesis of autoimmune diseases [9,10,11,12,13,14,15]. Epigenetic changes affect gene transcription by modulating the chromatin structure without altering the DNA sequence itself [16]. Epigenetic regulation contributes to the maintenance of a normal immune response. Previous reports have demonstrated that histone modifications are associated with the differentiation and function of T cells [17,18,19,20,21]. Revealing the epigenetic dysregulation in immune cells is important for understanding the pathogenesis of SLE and ultimately developing new therapeutic strategies (Figure 1).

## 2. The Pathogenesis of SLE

A line of evidence has shown that dysregulated adaptive and innate immune responses are closely associated with the pathogenesis of SLE [22]. SLE is an autoimmune disease that is caused by a breakdown in immunological self-tolerance. On the other hand, recent advances have demonstrated that SLE is a type I interferonopathy. To understand the complex pathogenesis of SLE, we review adaptive and innate immune responses in SLE.

### 2.1. Adaptive Immune Responses in SLE

The adaptive immune system attacks non-self-antigens but not self-antigens, which is referred to as self-tolerance. A breakdown in immunological self-tolerance leads to autoreactive immune responses through the induction of autoreactive lymphocytes, autoantibodies, and the impaired immunosuppressive function of regulatory T cells (Treg) [23,24,25]. The dysregulated adaptive immune responses attack the individual’s own tissues that include self-antigens. Immunological self-tolerance includes central and peripheral tolerance. In primary lymphoid tissues, central tolerance gets rid of autoreactive lymphocytes through apoptosis and receptor editing in B cells and through positive and negative selections in T cells [26,27,28,29]. In secondary lymphoid tissues, peripheral tolerance excludes autoreactive lymphocytes through anergy and follicular exclusion in B cells and through anergy, deletion, and suppression in T cells [30,31,32,33,34].

T follicular helper (Tfh) cells are programmed death receptor-1 (PD-1)^+^C-X-C motif chemokine receptor 5 (CXCR5)^+^CD4^+^ T cells that are involved in B cell differentiation and maturation in germinal centers of secondary lymphoid tissues [35]. Interleukin-12 (IL-12), IL-23, and transforming growth factor-β (TGF-β) induce the expression of B-cell/CLL lymphoma 6 (BCL6) and musculoaponeurotic fibrosarcoma (MAF) for Tfh cell differentiation. Tfh cells produce IL-21 and C-X-C motif chemokine ligand 13 (CXCL13), and play an important role in immunoglobulin class switching and affinity maturation in germinal centers. Tfh cells promote pathogenic B cell responses in SLE [36]. T peripheral helper (Tph) cells are PD-1^high^CXCR5^−^CD4^+^ T cells that are associated with B cell differentiation and tissue damage in peripheral inflammatory non-lymphoid tissues [37]. IL-12, TGF-β, IFNα, and IFNλ induce the expression of B lymphocyte-induced maturation protein-1 (BLIMP1) and MAF for Tph cell differentiation. Tph cells express C-C motif chemokine receptor 2 (CCR2), C-X3-C motif chemokine receptor 1 (CX3CR1), and CCR5, and produce IL-21 and CXCL13, similar to Tfh cells. Tph cells accumulate in peripheral inflamed tissues and induce pathogenic B cell responses in SLE [38]. Age-associated B cells (ABCs) are T-box transcription factor 21 (TBX21, also called T-bet)^+^CD11c^+^CD11b^+^CD21^−^ memory B cells that arise with age and are expanded in patients with autoimmune diseases such as SLE [39]. ABCs produce autoantibodies; secrete various cytokines, including IFNγ, tumor necrosis factor α (TNFα), and IL-17; and activate pathogenic T cells as antigen-presenting cells [40,41].

### 2.2. Innate Immune Responses in SLE

Activation of the innate immune system regulates adaptive immune responses [42]. Innate immune responses have been shown to be closely involved in the pathogenesis of SLE. Both Toll-like receptor (TLR)-dependent and TLR-independent innate immune pathways can induce type I IFN production in SLE. A broad expression of type I IFN-inducible genes, referred to as the IFN signature, has been shown in SLE [43]. In the TLR-dependent pathway, single-stranded RNAs or unmethylated CpG-rich double-stranded DNAs-containing immune complexes access TLR7 or TLR9, respectively, with the help of Fc receptors. Activation of the endosomal TLRs enhances IFNα production through interferon regulatory factor (IRF)5 and IRF7 in plasmacytoid dendritic cells (pDCs) [44]. In the TLR-independent pathways, small RNAs bind to intracytoplasmic RNA sensors, such as retinoic acid inducible gene-I (RIG-I) and melanoma differentiation-associated gene 5 (MDA5), and activate mitochondrial antiviral signaling (MAVS) [45]. Small RNAs also increase the permeability of mitochondria and release oxidized mitochondrial DNA into the cytosol [46]. Cytosolic DNA is associated with the DNA sensor cyclic GMP-AMP synthase (cGAS) and activates the stimulator of interferon genes (STING). These TLR-independent pathways induce IFNβ expression through IRF3 in macrophages.

Neutrophil extracellular traps (NETs) are mesh-like structures that are extruded from activated neutrophils in response to inflammatory stimuli [47]. NETs are composed of decondensed chromatin and intracellular proteins, including high mobility group box 1 (HMGB1), LL37 (a proteolytic fragment of cathelicidin), neutrophil elastase, and myeloperoxidase (MPO), which serve as autoantigens. NETs also include nucleic acids that activate the TLR pathway and the cGAS–STING pathway, resulting in the activation of innate immunity. NET formation, referred to as NETosis, contributes to the development of SLE [48].

## 3. Epigenetic Regulation of Chromatin Structure and Gene Transcription

Conrad H. Waddington proposed the epigenetic landscape theory that explains a process in which gene regulation modulates development. Epigenetic mechanisms regulate a stably heritable phenotype resulting from changes in a chromosome without alterations in the DNA sequence [49]. Epigenetic changes are conveyed through either mitosis or meiosis and modulate the chromatin structure, resulting in the change to gene transcription [50]. Chromatin is a mixture of DNA and proteins such as histones (H2A, H2B, H3, and H4), and forms chromosomes in the nucleus of eukaryotic cells [16,51]. DNA wraps around the histone proteins and forms the nucleosome that is the fundamental subunit of chromatin. The chromatin structure in regulatory regions of genomic DNA, including promoters and enhancers, alters the accessibility for transcription factors (TFs) [52]. In euchromatin, which is an accessible chromatin state, TFs are associated with genomic DNA and genes are actively transcribed [53]. In heterochromatin, which is a condensed chromatin state, TFs are not associated with genomic DNA and gene transcription is repressed.

DNA methylation means methylation of the fifth position of cytosine (5mC) [54]. DNA is methylated predominantly at the dinucleotide CpG in the promoters of genes [55]. A high degree of DNA methylation suppresses gene transcription by inhibiting the association of DNA and TFs [56]. DNA is methylated by DNA methyltransferases (DNMTs), such as de novo methyltransferases (DNMT3A, DNMT3B) and a maintenance methyltransferase DNMT1 [57,58]. Methyl-CpG-binding proteins (MBPs) that recognize methylated DNA comprise the three structural families [59]. A methyl-CpG-binding domain (MBD) protein family consists of MBD1, MBD2, MBD4, and methyl-CpG-binding protein 2 (MeCP2). DNMT1 maintains DNA methylation through an association with MeCP2 [60]. A SET and RING finger-associated (SRA) domain protein family includes ubiquitin-like with PHD and ring finger domains (UHRF)1 and UHRF2. UHRF1 binds to hemi-methylated DNA and maintains DNA methylation by recruiting DNMT1 [61]. A zinc finger protein family comprises Kaiso and Kaiso-like proteins. DNA demethylation is induced by a decrease in DNMT activity during cell division or by the ten-eleven translocation (TET) family of enzymes (TET1, TET2, and TET3).

Strahl and Allis proposed the histone code hypothesis that multiple histone modifications exhibit specific functions in a combinatorial or sequential fashion [62]. Covalent post-translational modifications in histone N-terminal tails, including acetylation and methylation, function as transcriptionally active or repressive markers [52,63,64]. Active histone markers, such as the acetylation of histone H3 and the methylation of histone H3 at lysine 4 (H3K4), are located on euchromatin and associated with active gene transcription. Repressive histone markers, such as methylation of H3K9 and H3K27, are located on heterochromatin and associated with repressive gene transcription. Histones are acetylated by histone acetyltransferases (HATs) and deacetylated by histone deacetylases (HDACs) [65]. Histones are methylated by histone methyltransferases (HMTs) and demethylated by histone demethylases (HDMs) [66].

miRNAs are a family of approximately 21-nucleotide-long small noncoding RNAs (ncRNAs) that are post-transcriptional regulators [67]. After miRNAs are associated with the 3′-untranslated region of messenger RNAs (mRNAs) of target genes, the perfect complementarity between miRNAs and its targets causes mRNA cleavage and imperfect complementarity induces translational repression [68,69]. Long ncRNAs (lncRNAs) are defined as >200-nucleotide-long ncRNAs and have recently gained attention. lncRNAs govern gene expression and are implicated in the pathogenesis of rheumatic diseases such as SLE [70].

## 4. Epigenetic Dysregulation in SLE

### 4.1. Dysregulated DNA Methylation in SLE

The dysregulation of DNA methylation has been demonstrated to relate with aberrant immune systems in SLE (Table 1). CD11a/CD18, which is also called lymphocyte function-associated antigen 1 (LFA-1) or integrin subunit alpha L (ITGAL), is an integrin that regulates leukocyte adhesion and migration in inflamed tissues. CD11a/CD18 overexpression correlates with the development of T cell autoreactivity [71]. The adoptive transfer of CD11a/CD18-overexpressed T cells into syngeneic mice caused a lupus-like disease [72]. DNA hypomethylation in the CD11a/CD18 promoter was identified in T cells from patients with active SLE [73]. CD70 (TNFSF7) is a ligand for CD27 and plays a role in T cell activation [74]. DNA hypomethylation in the CD70 promoter caused CD70 overexpression in SLE CD4^+^ T cells and induced B cell stimulation in SLE patients [75]. The CD40 ligand (CD40LG) that is expressed on activated CD4^+^ T cells, such as Tfh, functions as a costimulatory molecule and promotes B cell maturation [76]. The DNA hypomethylation of CD40LG contributes to CD40LG overexpression in CD4^+^ T cells from women with SLE [77]. The expression and activity of the transcription factor regulatory factor X-box 1 (RFX1) are repressed in SLE CD4^+^ T cells [78]. As RFX1 recruits DNMT1 and induces DNA methylation, the downregulated expression of RFX1 enhances CD11a/CD18 and CD70 expression in SLE CD4^+^ T cells. IL-10, which has B cell-promoting effects is elevated in the serum and tissues from SLE patients and induces autoantibodies by B cells [79]. DNA hypomethylation in the IL-10 gene and the recruitment of signal transducer and activator of transcription 3 (STAT3) to the IL-10 promoter and enhancer increase IL-10 expression in SLE T cells. Growth arrest and DNA damage-inducible 45 α (Gadd45α), which is a nuclear protein associated with the maintenance of genomic stability, DNA repair, and suppression of cell growth, plays a role in DNA demethylation [80]. Increased Gadd45α gene expression and global DNA hypomethylation enhanced CD11a/CD18 and CD70 gene expression in SLE CD4^+^ T cells [81]. Moreover, HMGB1 is associated with Gadd45α in CD4^+^ T cells during SLE flare [82]. HMGB1 expression is increased in SLE CD4^+^ T cells and associated with not only CD11a/CD18 and CD70 expression, but also the SLE Disease Activity Index (SLEDAI) score.

Eighty-six differentially methylated CpG sites in 47 genes were demonstrated in naïve CD4^+^ T cells from SLE patients by a genome-wide DNA methylation study [83]. Significant DNA hypomethylation is observed in type I IFN-regulated genes, such as MX dynamin-like GTPase 1 (MX1), bone marrow stromal cell antigen 2 (BST2), STAT1, tripartite motif containing 22 (TRIM22), IFN-induced proteins with tetratricopeptide repeats 1 (IFIT1), IFIT3, IFN-induced protein 44 like (IFI44L), and ubiquitin specific peptidase 18 (USP18). Another group reported differentially methylated genes, including sphingosine-1-phosphate receptor 3 (S1PR3), CD300 molecule-like family member B (CD300LB), and NACHT, LRR, and PYD domains-containing protein 2 (NLRP2) in SLE CD4^+^ T cells using a genome-wide DNA methylation experiment [84]. A decrease in signaling through the RAS-mitogen-activated protein kinase (MAPK) pathway contributes to decreased DNMT activity and DNA hypomethylation in T cells from patients with active SLE [85]. Enhanced levels of the catalytic subunit of protein phosphatase 2A (PP2Ac) inhibit the mitogen-activated protein kinase kinase (MEK)/extracellular signal-regulated kinase (ERK) signaling pathway and decreases DNMT1 expression and DNA methylation in SLE T cells [86]. An increased expression of miR-21 and miR-148a decreases the DNMT1 expression and induces DNA hypomethylation, resulting in the overexpression of CD11a/CD18 and CD70 genes in SLE CD4^+^ T cells [87]. IL-17A induces the production of cytokines and chemokines, such as IL-6 and IL-8, and recruits monocytes and neutrophils. IL-17A is implicated in the development of SLE [88]. Decreased DNMT3A and HDAC1 causes DNA hypomethylation and H3K18 hyperacetylation in the IL-17A gene, respectively, in SLE T cells [89]. The cAMP-responsive element modulator α (CREMα) is highly expressed and associated with the IL-17A promoter, resulting in an increased IL-17A production in SLE T cells. An epigenome-wide study revealed that differential methylated genes that regulate the response to tissue hypoxia and IFN-mediated signaling contributed to lupus nephritis [90]. The downregulation of MBD4 decreased the DNA methylation of the CD70 gene and enhanced CD70 gene expression in SLE CD4^+^ T cells [91]. Decreased 3-hydroxy butyrate dehydrogenase 2 (BDH2), a modulator of intracellular iron homeostasis, leads to DNA hypomethylation via increasing the amount of intracellular iron in SLE CD4^+^ T cells [92]. 

DNA hypomethylation in IFN-associated genes, such as poly (ADP-ribose) polymerase family member 9 (PARP9), IFN-induced transmembrane protein 1 (IFITM1), and IFI44L was demonstrated in CD4^+^ T cells, B cells, granulocytes, and monocytes from SLE patients [93]. Differentially methylated genes, such as MX1, IFI44L, IFIT1, IFI44, IRF5, IRF7, MHC-class III, PARP9, UHRF1-binding protein 1 (UHRF1BP1), radical S-adenosyl methionine domain containing 2 (RSAD2), phospholipid scramblase 1 (PLSCR1), ubiquitin-conjugating enzyme E2 L3 (UBE2L3), CD45, Ikaros family zinc finger 3 (IKZF3), and histone E3 ubiquitin ligase 3L (DT3XL), were identified in peripheral blood mononuclear cells (PBMCs) from SLE patients [94,95]. An analysis of DNA methylation profiles identified hypomethylated genes, such as IFI44, IFITM1, Y-box binding protein 1 (YBX1), and TATA-box binding protein associated factor 8 (TAF8), and hypermethylated genes, including SRY-box transcription factor 12 (SOX12), ADP-ribosylation factor GTPase-activating protein 3 (ARFGAP3), coiled-coil domain containing 81 (CCDC81), and maternally expressed 3 (MEG3), in SLE B cells [96]. Downregulated UHRF1 enhances the BLC6 expression through an decrease in DNA methylation and trimethylation at H3K27 (H3K27me3) levels in the BCL6 promoter and promotes Tfh cell differentiation in SLE [97].

### 4.2. Dysregulated Histone Modifications in SLE

In this section, we review the roles of dysregulated histone modifications in aberrant immune responses in SLE (Table 2). In PBMCs from SLE patients, the levels of H3K4me3 are increased in WD repeat-containing protein 5 (WDR5) and solute carrier family 24 member 3 (SLC24A3) genes, and decreased in protein tyrosine phosphatase non-receptor type 22 (PTPN22), methyltransferase 16 (METTL16), LDL receptor-related protein 1B (LRP1B), and cadherin 13 (CDH13) genes [98]. CD70 gene expression is increased in CD4^+^ T cells from active SLE patients [99]. Acetylation at histone H3 (H3ac) and dimethylation at H3K4 (H3K4me2) levels in the CD70 gene are enhanced in SLE CD4^+^ T cells. TNFα expressing monocytes were more frequent in SLE patients compared to healthy controls. H3ac levels were not increased but H4ac levels were increased in the TNFα gene in SLE monocytes [100]. The increased expression of PP2Ac contributes to the production of IL-17 by enhancing H3ac through the activation of IRF4 in SLE T cells [101]. A genome-wide analysis revealed that H4ac levels are increased in the genes that are regulated by IRF1 in SLE monocytes [102]. H3K27me3 is increased in the hematopoietic progenitor kinase 1 (HPK1, also called MAP4K1) gene in SLE CD4^+^ T cells. As a result, downregulating HPK1 induces T cell activation [103]. Enhanced H3K27me3 enrichment in the HPK1 promoter is caused by a decrease in Jumonji domain-containing protein 3 (JMJD3) binding in SLE CD4^+^ T cells. Global histones H3 and H4 hypoacetylation was observed in CD4^+^ T cells from active SLE patients [104]. The degree of H3ac was inversely associated with the SLEDAI score. Global H3K9 hypomethylation was demonstrated in CD4^+^ T cells from both active and inactive SLE patients. However, global levels of H3K4 methylation were similar in CD4^+^ T cells from SLE patients compared to healthy controls. Sirtuin 1 (SIRT1) mRNA levels were enhanced, while mRNA levels of HDAC2, HDAC7, P300, cyclic AMP response element-binding protein (CBP), enhancer of zeste homolog 2 (EZH2), and suppressor of variegation 3-9 homolog 2 (SUV39H2) were reduced in CD4^+^ T cells from active SLE patients. 

RFX1 decreases the levels of H3ac and increases those of H3K9me3 by recruiting HDAC1 and SUV39H1, respectively [78,105]. Therefore, the downregulated expression of RFX1 enhances CD11a/CD18 and CD70 expression in SLE CD4^+^ T cells. Overexpressed CREMα suppresses IL-2 expression through HDAC1-mediated H3K18 deacetylation and DNMT3A-mediated DNA hypermethylation in SLE T cells [106]. Reduced IL-2 expression contributes to the decrease in Treg in SLE patients. Trichostatin A (TSA), an inhibitor of HDAC, impairs the increased expression of CD40LG and IL-10 genes, and decreased IFNγ gene expression in SLE CD4^+^ T cells [107]. TLR2 expression was enhanced in CD4^+^ T cells, CD8^+^ T cells, B cells, and monocytes from SLE patients [108]. In SLE CD4^+^ T cells, TLR2 stimulation increased CD40LG, CD70, IL-6, IL-17A, IL-17F, and TNFα expression and decreased the expression of forkhead box P3 (FOXP3), which is a master regulator of Treg. TLR2 activation enhanced H4ac levels and reduced H3K9me3 levels in the IL-17A and IL-17F genes in SLE CD4^+^ T cells. TNFα-induced protein 3 (TNFAIP3) expression was downregulated by decreasing the H3K4me3 level in the gene promoter in SLE CD4^+^ T cells [109]. The reduced expression of TNFAIP3 increased the expression of IFNγ and IL-17. BCL6 was highly expressed and repressed the miR-142-3p/5p expression by increasing H3K27me3 levels and decreasing H3K9/14ac levels in SLE CD4^+^ T cells [110]. BCL6 recruited EZH2 and HDAC5 to the miR-142-3p/5p promoter and induced the expression of IL-21, inducible T-cell co-stimulator (ICOS), and CD40LG. BCL-6, IL-21, ICOS, and CD40LG play an important role in the development and function of Tfh cells, resulting in CD4^+^ T cell hyperactivity and autoantibody production in SLE. IL-23 promoted the phosphorylation of STAT3 in T helper 17 (Th17) cells, which are a subset of CD4^+^ T helper cells defined by the production of IL-17, from SLE patients [111]. IL-23 enhanced the H3K4me3 level and reduced the H3K27me3 level in the retinoic acid receptor-related orphan receptor γt (RORγt) gene, which is a master regulator of Th17 cells. IL-23-induced STAT3 binds to the RORγt gene locus, leading to an increase in RORγt expression.

### 4.3. Dysregulated miRNA Expression in SLE

Dysregulated miRNA expression contributes to the immunopathogenesis of SLE (Table 3). Estrogen-induced miR-10b-5p is increased in SLE T cells [112]. miR-10b-5p targets serine/arginine-rich splicing factor 1 (SRSF1), which promotes IL-2 production. Therefore, the decrease in SRSF1 impairs Treg differentiation and leads to the persistence of autoreactive T cells in SLE [113]. T cell-restricted *Srsf1*-deficient mice showed high frequencies of activated T cells producing proinflammatory cytokines and systemic autoimmunity such as lupus-like nephritis [114]. miR-17-5p expression is repressed by E2F transcription factor 2 (E2F2) in SLE B cells [115]. miR-17-5p targets IL-10, which has B cell-promoting effects, resulting in autoantibody production. miR-21 expression is elevated in SLE CD4^+^ T cells and associated with SLEDAI score [116]. miR-21 targets programmed cell death protein 4 (PDCD4), which suppresses proliferation, IL-10 production, CD40LG expression, and the capacity to drive B cell maturation in SLE CD4^+^ T cells [117,118,119,120]. miR-21 also targets BDH2, which promotes DNA methylation in SLE CD4^+^ T cells [92]. IL-6 or TNFα activates NF-kB/p65, which binds to the miR-34a promoter and increases miR-34a expression in the PBMC and CD4^+^ T cells from SLE patients [121]. miR-34a targets FOXP3, which promotes the development of Treg. miR-98 expression is repressed in SLE CD4^+^ T cells [122]. miR-98 targets FAS, which promotes the apoptotic signaling pathway. miR-99a-3p expression is downregulated in SLE PBMCs [123]. miR-99a-3p targets eukaryotic translation initiation factor 4E-binding protein 1 (EIF4EBP1), which promotes the autophagy signaling pathway in B cells.

miR-125a expression is downregulated in SLE T cells [124]. miR-125a targets Kruppel-like factor 13 (KLF13), which enhances regulated activation, normal T cell expressed, and secreted (RANTES) expression. miR-142-3p expression is increased in SLE monocyte-derived DCs, which produces C-C motif chemokine ligand 2 (CCL2), CCL5, CXCL8, IL-6, and TNFα [125]. Monocyte-derived DCs attract CD4^+^ T cells and suppress the production of Treg in SLE. miR-146a expression is downregulated in SLE PBMCs [126,127]. miR-146a is a negative regulator of the type I IFN signaling pathway and is inversely correlated with the SLEDAI score and the IFN score in SLE patients. miR-146a targets IRF5 and STAT1. miR-146a expression is upregulated in SLE Treg [128]. miR-146a targets STAT1, which promotes the Th1 response and is associated with autoimmunity. miR-152-3p expression is increased in SLE B cells [129]. miR-152-3p targets KLF5, which reduces BAFF expression. miR-152-3p expression was elevated in the CD4^+^ T cells and PBMCs from SLE patients [130]. An increased miR-152-3p expression is correlated with skin rashes, arthritis, anti-dsDNA antibody, and IgG in SLE patients. miR-152-3p induces the autoreactivity of CD4^+^ T cells and TLR-mediated inflammatory responses by targeting DNMT1, which increases DNA methylation in the myeloid differentiation factor 88 (MyD88) gene. miR-155 expression is repressed in serum, urine, and PBMCs from SLE patients [131,132]. miR-155 targets CAMP response element binding protein (CREB), which promotes PP2A expression. As PP2A inhibits IL-2 production, downregulated miR-155 suppresses IL-2 production in SLE PBMCs. miR-155 expression is negatively correlated with the SLEDAI score and proteinuria, and is positively correlated with white blood cell counts in SLE patients [132].

Estrogen-regulated miR-302d expression was reduced in SLE monocytes [133]. miR-302d targets IRF9, which is involved in the expression of IFN-stimulated genes such as 2′-5′-oligoadenylate synthetase 1 (OAS1) and MX1. miR-302d expression is negatively correlated with IFN score in SLE patients. miR-663 expression is increased in the bone marrow-derived mesenchymal stem cells (BMSCs) from SLE patients [134]. BMSCs inhibit the proliferation and function of immune cells, such as T cells, B cells, natural killer cells, and DCs. miR-663 reduces the proliferation and migration of BMSCs and suppresses the BMSC-mediated decrease in Tfh cells and increase in Treg by targeting TGF-β in SLE patients. miR-663 is associated with the SLEDAI score. miR-4512 expression is decreased in monocytes and macrophages from SLE patients [135]. As miR-4512 targets TLR4, NETosis is induced by the activation of the TLR4 pathway in SLE patients.

lncRNAs are another highly diverse class of ncRNAs that are involved in genomic, transcriptional, and translational regulation of their target genes [136]. A variety of lncRNAs, including growth arrest-specific transcript 5 (GAS5), nuclear paraspeckle assembly transcript 1 (NEAT1), and metastasis-associated lung adenocarcinoma transcript 1 (MALAT1), are implicated in the pathogenesis of SLE [137,138,139].

**Table 1 ijms-25-01019-t001:** Aberrant DNA methylation in SLE. This table lists DNA methylation states, effects by aberrant DNA methylation, and types of experiments and cells that were examined for DNA methylation evaluation in SLE-associated genes.

Genes	DNA Methylation	Effects	Types of Experiments	Cells	Refs.
CD11a/CD18	hypomethylation	leukocyte adhesion and migration in inflamed tissues	in vitro	T cells	[73]
CD11a/CD18	hypomethylation	leukocyte adhesion and migration in inflamed tissues	in vitro	CD4^+^ T cells	[78,81,87]
CD70	hypomethylation	B cell activation	in vitro	CD4^+^ T cells	[75,78,81,87,91]
CD40LG	hypomethylation	B cell maturation	in vitro	CD4^+^ T cells	[77]
IL-10	hypomethylation	B cell activation	in vitro	T cells	[79]
MX1, BST2, STAT1, TRIM22, IFIT1, IFIT3, IFI44L, USP18	hypomethylation	type I IFN-mediated responses	in vitro	naïve CD4^+^ T cells	[83]
S1PR3, CD300LB, NLRP2	hypomethylation		in vitro	CD4^+^ T cells	[84]
IL-17A	hypomethylation	recruitment of monocytes and neutrophils	in vitro	T cells	[89]
PARP9, IFITM1, IFI44L	hypomethylation	type I IFN-mediated responses	in vitro	CD4^+^ T cells, B cells, granulocytes, monocytes	[93]
MX1, IFI44L, PARP9, DT3XL, IFIT1, IFI44, RSAD2, PLSCR1, IRF7	hypomethylation	type I IFN-mediated responses	in vitro	PBMCs	[95]
IFI44, IFITM1, YBX1, TAF8	hypomethylation		in vitro	B cells	[96]
SOX12, ARFGAP3, CCDC81, MEG3	hypermethylation		in vitro	B cells	[96]
BCL6	hypomethylation	Tfh cell differentiation	in vitro	naïve CD4^+^ T cells	[97]
IL-2	hypermethylation	Treg suppression	in vitro	T cells	[106]

**Table 2 ijms-25-01019-t002:** Aberrant histone modifications in SLE. This table lists histone modification states, effects by aberrant histone modifications, and types of experiments and cells that were examined for histone modification evaluation in SLE-associated genes.

Genes	Histone Modifications	Effects	Types of Experiments	Cells	Refs.
WDR5, SLC24A3	high H3K4me3		in vitro	PBMCs	[98]
PTPN22, METTL16, LRP1B, CDH13	low H3K4me3		in vitro	PBMCs	[98]
CD70	high H3K4me2, high H3ac	B cell activation	in vitro	CD4^+^ T cells	[99]
TNFα	high H4ac	inflammatory responses	in vitro	monocytes	[100]
IL-17	high H3ac	recruitment of monocytes and neutrophils	in vitro	T cells	[101]
IL-17A, IL-17F	high H4ac, low H3K9me3	recruitment of monocytes and neutrophils	in vitro	CD4^+^ T cells	[108]
HPK1	high H3K27me3	T cell activation	in vitro	CD4^+^ T cells	[103]
CD11a/CD18	low H3K9me3, high H3ac	leukocyte adhesion and migration in inflamed tissues	in vitro	CD4^+^ T cells	[78,105]
CD70	low H3K9me3, high H3ac	T cell activation	in vitro	CD4^+^ T cells	[78,105]
IL-2	low H3K18ac	Treg suppression	in vitro	T cells	[106]
TNFAIP3	low H3K4me3	production of IFNγ and IL-17	in vitro	CD4^+^ T cells	[109]
miR-142-3p/5p	high H3K27me3, low H3K9/14ac	production of IL-21, ICOS, and CD40LG, and Tfh cell differentiation	in vitro	CD4^+^ T cells	[110]
RORγt	high H3K4me3, low H3K27me3	Th17 differentiation	in vitro	CD4^+^ T cells	[111]

**Table 3 ijms-25-01019-t003:** Aberrant miRNA expression in SLE. This table lists miRNA expression, effects by aberrant miRNA expression, and types of experiments and cells that were examined for miRNA expression in SLE-associated genes.

miRNA Expression	Target Genes	Effects	Types of Experiments	Cells	Refs.
miR-10b-5p upregulation	SRSF1	Treg suppression	in vitro	T cells	[112]
miR-17-5p downregulation	IL-10	autoantibody production	in vitro	B cells	[115]
miR-21 upregulation	PDCD4	cell proliferation and B cell maturation	in vitro	CD4^+^ T cells	[116]
miR-21 upregulation	BDH2	suppression of DNA methylation	in vitro	CD4^+^ T cells	[92]
miR-34a upregulation	FOXP3	Treg suppression	in vitro	PBMCs, CD4^+^ T cells	[121]
miR-98 downregulation	FAS	apoptosis	in vitro	CD4^+^ T cells	[122]
miR-99a-3p downregulation	EIF4EBP1	autophagy	in vitro	PBMCs	[123]
miR-125a downregulation	KLF13	increase in RANTES expression	in vitro	T cells	[124]
miR-146a downregulation	STAT1	Th1 responses	in vitro	PBMCs	[126,127]
miR-152-3p upregulation	KLF5	increase in BAFF expression	in vitro	B cells	[129]
miR-152-3p upregulation	DNMT1	autoreactive CD4^+^ T cell responses and TLR-mediated inflammatory responses	in vitro	PBMCs, CD4^+^ T cells	[130]
miR-155 downregulation	CREB	reduced IL-2 production	in vitro	PBMCs	[131,132]
miR-302d downregulation	IRF9	type I IFN-mediated responses	in vitro	monocytes	[133]
miR-663 upregulation	TGF-β1	Tfh cell activation and Treg suppression	in vitro	BMSCs	[134]
miR4512 downregulation	TLR	NETosis	in vitro	monocytes, macrophages	[135]

## 5. Conclusions

SLE is a systemic autoimmune disease characterized by autoreactive immune responses and a type I IFN signature. Epigenetic mechanisms regulate gene transcription by modulating the chromatin structure. Although genetic and environmental factors have been shown to be involved in the pathogenesis of SLE, increasing evidence has demonstrated that dysregulated epigenetic changes contribute to aberrant immune responses, resulting in the progression of SLE. From the perspective of adaptive immunity, dysregulated epigenetic mechanisms play a critical role in the activation of T cells in SLE. Activated Tfh cells promote autoantibody production. Epigenetic dysregulation also suppresses Treg differentiation and increases the number of autoreactive lymphocytes. Regarding innate immunity, dysregulated epigenetic changes promote type I IFN production that activates type I IFN-regulated genes in SLE. Thus, the complex pathogenesis of SLE has been unraveled. Recently, the development of genome-wide analyses using next-generation sequencing has minutely elucidated the roles of epigenetic mechanisms as well as the molecules that influence epigenetic changes in SLE. These analyses will facilitate the development of novel drugs targeting the dysregulated epigenetic changes. Advances in our understanding of the roles of epigenetic dysregulation in SLE will shed further light on the pathogenesis of SLE and pave the way for discovering new therapeutic strategies and biomarkers for SLE.

## Figures and Tables

**Figure 1 ijms-25-01019-f001:**
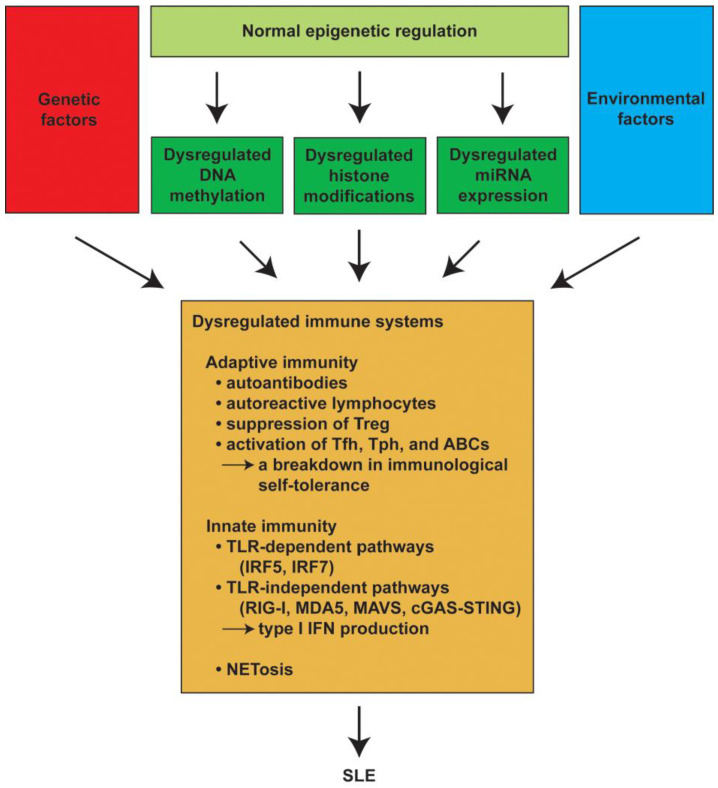
Genetic factors, environmental factors, and dysregurated epigenetic mechanisms, such as DNA methylation, histone modifications, and miRNA expression, induce aberrant immune systems, including adaptive immunity and innate immunity, resulting in the development of SLE. SLE: systemic lupus erythematosus; miRNA: microRNA; Treg: regulatory T cells; Tfh: T follicular helper cells; Tph: T peripheral helper cells; ABCs: age-associated B cells; TLR: Toll-like receptor; IRF: interferon regulatory factor; RIG-I: retinoic acid inducible gene-I; MDA5: melanoma differentiation-associated gene 5; MAVS: mitochondrial antiviral signaling; cGAS: cyclic GMP-AMP synthase; STING: stimulator of interferon genes; IFN: interferon.

## Data Availability

Not applicable.

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
