# Peer review of "Epigenetic Dysregulation in the Pathogenesis of Systemic Lupus Erythematosus"

_ijms, 2024, doi:10.3390/ijms25021019_

Round 1

Reviewer 1 Report

Comments and Suggestions for Authors

The authors described the epigenetic modifications involved in the pathogenesis of SLE.

The topic is not absolutely original as others have evaluated the role of epigenetic modifications in SLE but in this work a lot of new information has been provided.

Compared to other publications with the same topics, the authors have collected different information on Epigenetic regulation of chromatin structure and gene transcription and have listed in the various tables all the genes with abnormalities of DNA methylation, histone modifications and miRNA expression.

As a review, the tables are appropriate and well done with the list of all the genes and their references.

The conclusions are short but appropriate.

As review it is ok in the present form.

Author Response

We appreciate your review very much.

Reviewer 2 Report

Comments and Suggestions for Authors

Epigenetic dysregulation in the pathogenesis of systemic lupus erythematosus

Systemic lupus erythematosus (SLE) is a chronic autoimmune pathology with complex pathogenic mechanisms. Dysregulated DNA methylation, histone modifications and microRNA expression are notably involved in the pathogenesis of SLE, with participation in disease manifestations. As such, miR21-expression, miR-663, among others, are associated with SLEDAI score, while, for example, miR-155 expression is negatively correlated with SLEDAI score and proteinuria, and positively correlated with white blood cell count. Therefore, the subject of this review is relevant, both as an analysis of the cumulated knowledge on the subject of SLE pathogenic mechanisms regarding epigenetic dysregulation, evaluating its association to clinical aspects, such as disease activity score and manifestations, but also as a reflection of the importance of such knowledge for further improving therapeutic options in SLE, and as an impetus for achieving this goal. The review emphasizes the need for further study on this aspect of pathogenesis to allow for a continuing expansion of treatment possibilities in such a debilitating disease as SLE.

The manuscript is well-structured and logically organized, which are essential elements, given the complexity and difficulty of the subject. The high level of written clarity and fluency of ideas are obvious throughout the review and the text is concise, rendering the manuscript easy to comprehend. The scientific content is of optimal quality, with a gradual progression of the difficulty of notions, each relevant concept is well described and explained, while also being thoroughly referenced and rigorously researched. The use of tables that succinctly present the information previously elaborated, further allows for a greater degree of readability.

After analyzing this manuscript, it can be considered for publication.

Author Response

We appreciate your review very much.

Reviewer 3 Report

Comments and Suggestions for Authors

In this review article, Drs. Araki and Mimura discussed the epigenetic alteration in SLE. Overall, the manuscript is nicely written, however, I do have some concerns with the topic:

1) This is not a novel and new topic to be discussed. There are so many reviews out there discussing the same topic (PMID: 28752494; 28752494; 27396525 and so on). Thus, why do we need another one? 

2) "Disregulated histone modifications in SLE (Table 2)" Please do not add the bookmark of table 2 in the heading. Please also revise the others

3) These reviews also discuss the (immuno)pathogenesis of SLE. Consider including (PMID: 37374237; 30796732)

4) The authors must include some figures as those would help readers to understand the context. For instance, the authors could compare the physiological epigenetics and pathological ones in SLE.

5) In table 3, what are the consequences of those up/downregulation of miRNA? Please add

6) Since lncRNA was not discussed, I guess there is no alteration in SLE? 

Comments on the Quality of English Language

No comment

Author Response

We would like to thank you for your detailed comments and suggestions. This manuscript has now been extensively revised in line with these suggestions. All of the changes in our revised manuscript have been highlighted in a red font. Our point-by-point responses are listed below.

Comment 1: This is not a novel and new topic to be discussed. There are so many reviews out there discussing the same topic (PMID: 28752494; 28752494; 27396525 and so on). Thus, why do we need another one?

Response: As you pointed out, the role of epigenetic regulation in the pathogenesis of SLE have been well discussed. However, new findings are being reported in this field. In this paper, we reviewed the most recent findings and discussed the role of dysregulated epigenetic mechanisms in aberrant immune systems that are involved in the pathogenesis of SLE. We are convinced that we updated the knowledge in this field.

Comment 2: "Disregulated histone modifications in SLE (Table 2)" Please do not add the bookmark of table 2 in the heading. Please also revise the others

Response: We appreciate your good comment. We omitted the titles in Table 1-3.

Comment 3: These reviews also discuss the (immuno)pathogenesis of SLE. Consider including (PMID:37374237; 30796732)

Response: We agree with your suggestion. We Added the references and described the immunopathogenesis of SLE in our revised manuscript.

Comment 4: The authors must include some figures as those would help readers to understand the context. For instance, the authors could compare the physiological epigenetics and pathological ones in SLE.

Response: We appreciate your essential comment. We added a figure regarding the role of dysregulated epigenetic mechanisms in the pathogenesis of SLE in our revised manuscript.

Comment 5: In table 3, what are the consequences of those up/downregulation of miRNA? Please add

Response: This is a good point. We added the effects of up- or down-regulated miRNAs in the pathogenesis of SLE in Table 3.

Comment 6: Since lncRNA was not discussed, I guess there is no alteration in SLE?

Response: We appreciate your important comment. We described the role of lncRNA in the pathogenesis of SLE in our revised manuscript.

Round 2

Reviewer 3 Report

Comments and Suggestions for Authors

Thanks for sending the revision. I do still have some comments:

1) This typological issue still occurs "4. Dysregulated DNA methylation in SLE (Table 1)" please fix

2) I think section 4,5,6 should be the subsections of a new section somewhat called epigenetic dysregulation in SLE. Consider adding

3) Section 4 about DNA methylation is too long for a paragraph. Break it down into several paragraphs

4) Similarly, break section 5 and 6 into several paragraphs

5) "Table 1" what is this about? there is no title, no explanation at all. Please correct

6) Similarly, add the title and some explanations about table 2 and 3. 

7) Link those tables with the text

8) Table 1 is not informative. What are the effects of those hypomethylation? Seen in which type of experiments (in vitro, in vivo etc)? What was the model used (PBMC etc)? Please add

9) Similarly, add those information in Table 2 as well

10) "miR-10b-5p up SRSF1 T cell hyperactivity", which type of T cell? please specify

11) Similarly, add the study design in table 3, and add the model used in those experiments.

12) In the conclusion, "Increasing evidence has demonstrated that dysregulated epigenetic changes participate in the autoreactive immune responses, resulting in the progression of SLE", please specify what are those dysregulation specifically? conclusion should be a useful summary of the text. 

Comments on the Quality of English Language

No comment

Author Response

We would like to thank you for your detailed comments and suggestions. This manuscript has now been extensively revised in line with these suggestions. All of the changes in our revised manuscript have been highlighted in a red font. Our point-by-point responses are listed below.

Comment 1: This typological issue still occurs "4. Dysregulated DNA methylation in SLE (Table 1)" please fix

Response: We appreciate your good comment. We corrected the typological issue that you pointed out.

Comment 2: I think section 4,5,6 should be the subsections of a new section somewhat called epigenetic dysregulation in SLE. Consider adding

Response: We appreciate your good comment. We made a new section and three subsections you proposed.

Comment 3: Section 4 about DNA methylation is too long for a paragraph. Break it down into several paragraphs

Response: We appreciate your good comment. We broke down section 4 into three paragraphs.

Comment 4: Similarly, break section 5 and 6 into several paragraphs

Response: We appreciate your good comment. We broke down section 5 and 6 into several paragraphs.

Comment 5: "Table 1" what is this about? there is no title, no explanation at all. Please correct

Response: We appreciate your good comment. We added the title and explanation of Table 1.

Comment 6: Similarly, add the title and some explanations about table 2 and 3. 

Response: We appreciate your good comment. We added the titles and explanations of Table 2 and 3.

Comment 7: Link those tables with the text

Response: We appreciate your good comment. We linked Table 1, 2, and 3 with the text.

Comment 8: Table 1 is not informative. What are the effects of those hypomethylation? Seen in which type of experiments (in vitro, in vivo etc)? What was the model used (PBMC etc)? Please add

Response: We appreciate your important comment. We added the information about the effects, the type of experiments, and the model used in the experiments in Table 1.

Comment 9: Similarly, add those information in Table 2 as well

Response: We appreciate your important comment. We added the information about the effects, the type of experiments, and the model used in the experiments in Table 2.

Comment 10: "miR-10b-5p up SRSF1 T cell hyperactivity", which type of T cell? please specify

Response: We appreciate your important comment. As below, we described the effects of miR-10b-5p in SLE T cells in the text and corrected table 3. miR-10b-5p is increased in SLE T cells. Because miR-10b-5p targets SRSF1, SRSF1 expression is decreased in SLE T cells. SRSF1 was reported to promote IL-2 production (Moulton VR et al. Proc Natl Acad Sci USA. 2013;110:1845-50). As IL-2 induces Treg differentiation, the decrease in SRSF1 impairs Treg differentiation and leads to persistence of autoreactive T cells in SLE. In addition, T cell-restricted Srsf1-deficient mice showed high frequencies of activated T cells producing proinflammatory cytokines and systemic autoimmunity such as lupus-like nephritis (Katsuyama T et al. J Clin Invest. 2019;129:5411-23).

Comment 11: Similarly, add the study design in table 3, and add the model used in those experiments.

Response: We appreciate your important comment. We added the information about the type of experiments and the model used in the experiments in Table 3.

Comment 12: In the conclusion, "Increasing evidence has demonstrated that dysregulated epigenetic changes participate in the autoreactive immune responses, resulting in the progression of SLE", please specify what are those dysregulation specifically? conclusion should be a useful summary of the text. 

Response: We appreciate your important comment. We summarized the dysregulated epigenetic changes in the conclusion.

Round 3

Reviewer 3 Report

Comments and Suggestions for Authors

Improve the quality and format of those tables first. Table 1 is still messed up. Table 2 is not complete. Some of the effects are missing. Also, make sure that the references in those tables are linked to the reference list.

Comments on the Quality of English Language

no comment

Author Response

We would like to thank you for your detailed comments and suggestions. This manuscript has now been extensively revised in line with these suggestions. All of the changes in our revised manuscript have been highlighted in a red font. Our point-by-point responses are listed below.

Comment: Improve the quality and format of those tables first. Table 1 is still messed up. Table 2 is not complete. Some of the effects are missing. Also, make sure that the references in those tables are linked to the reference list.

Response: We appreciate your good comment. We corrected Table 1, Table 2, and Table 3. We rewrote the effects by epigenetic dysregulation and the types of experiments and cells that were examined for the evaluation of epigenetic mechanisms in SLE-associated genes in those tables. We also confirm that the references in those tables are linked to the reference list.